# “I Learned More Because I Became More Involved”: Teacher’s and Students’ Voice on Gamification in Physical Education Teacher Education

**DOI:** 10.3390/ijerph20043038

**Published:** 2023-02-09

**Authors:** Gonzalo Flores-Aguilar, María Prat-Grau, Jesús Fernández-Gavira, Antonio Muñoz-Llerena

**Affiliations:** 1Physical Education and Sport Department, Universidad de Sevilla, 41013 Seville, Spain; 2Research Group “Social Inclusion, Physical Education and Sport, and European Policies in Research” (INEFYD), Universidad de Sevilla, 41013 Seville, Spain; 3Sport Research Institute UAB, Universitat Autònoma de Barcelona, 08193 Bellaterra, Spain

**Keywords:** transformative learning, sustainable development goals, higher education, physical education, transference, innovation

## Abstract

Higher education plays a critical role in achieving the Sustainable Development Goals established in the 2030 Agenda, especially the fourth goal (quality and equality in higher education). Therefore, teacher education must play a central role in providing transformative learning experiences for future teachers that can lead the change to create high quality programs in every school. The purpose of this study was to conduct a gamified experience in Physical Education Teacher Education with two goals: assess the students’ views on the framework and evaluate the teachers’ feelings and thoughts. One teacher-researcher (36 years) and 74 students (19–27 years) enrolled in a Spanish university agreed to participate. A qualitative descriptive method and an action-research design were used. The teacher-researcher completed a personal diary, while the students answered two open-ended questions. From the students’ responses emerged three positive themes (framework, motivation, and transference) and two negatives (boredom and group work); from the teacher-researcher, we received three positive responses (mixed emotions, expectations, and students’ motivation) and one negative (workload). As a conclusion, gamification could be considered a framework that promotes transformative learning.

## 1. Introduction

Higher education plays a critical role in the achievement of the Sustainable Development Goals established in the Agenda 2030 by the United Nations, due to its impact on knowledge generation and dissemination within society [1], and it is explicitly included within Sustainable Development Goal 4 (i.e., quality and equality in education). Therefore, there is a call to renew teacher training programs to improve the teaching-learning process in compulsory education since teacher education can be a key element to assess and integrate new pedagogical approaches [2]. To promote this change, teacher education must play a central role in providing transformative learning experiences for future teachers who can lead the change to create high quality programs in every school [3]. Transformative learning begins when individuals “look at old things in new ways” and finishes when they “do new things in new ways” [4]. Four core elements have been identified in this framework [5]: (a) critical reflection: it goes beyond mere doing and is needed to examine personal values and/or beliefs; (b) dialogue: it helps individuals revise assumptions, putting critical reflection to work; (c) individual experience: first-person experiences are crucial, but they are culturally-biased, and only through critical reflection and dialogue can individuals uncover those biases; and (d) context: the time or the setting plays a significant role in the learning process.

Teachers (including higher education) must incorporate new methodological approaches that shift the focus from the teacher to the students [6] to provide successful experiences that can fulfill the students’ needs, increasing their motivation [7]. Teacher education should adopt a proactive role to assess and integrate new methodological approaches and upgrade physical education into the 21st century. The irruption of gamification in the different educational stages has increased over the last decade due to the rise in the use of games and mobile applications in these contexts [8]. Basically, it consists of the introduction of elements from games or video games in non-game contexts [9] with the intention of causing behavioral changes in the participants (players) due to the motivation that it generates [10]. It was originally designed to be used in business, but starting in 2008, it moved to other contexts, such as health, marketing, human resources, and, finally, education [11]. According to [9], game design elements can be grouped into three categories (Figure 1): (1) dynamics: they represent the highest conceptual level, portray a comprehensive view of the whole project, and include elements like the narrative, the emotions, the progressions, or the relationships; (2) mechanics: as the second conceptual level, they contain the basic elements that make the action in the game progress, such as rules, challenges, chance, rewards, competition, cooperation, feedback, or resource acquisition; and (3) components: they refer to the basic conceptual level and include more specific elements of the framework, such as avatars, teams, badges, achievements, levels, points, goods, or scoreboards.

Gamification has been found to increase students’ intrinsic motivation in different contexts [12]. However, it can promote positive outcomes only when it is properly structured [13]. According to [14], its effectiveness increases when the framework feeds the participants’ curiosity and uncertainty within a positive emotional climate (joy, fun). The implementation of gamified frameworks in educational contexts requires adding elements to the ones previously mentioned: (1) type of players: achievers, explorers, killers, or socializers [15]; (2) flow: state of concentration [16]; and (3) pedagogical orientations: didactic and gamified phases [17]. On the other hand, the use of mistaken frameworks in education, based on one-off experiences, rewards, and awards in a competitive context, searching only for fun without learning goals, can reduce its potential [10]. An incorrectly implemented gamification can lead to diminished benefits and even negative effects [18]. 

Researchers warned that the presence of gamification in the scientific scenario is still scarce [10]. A recent review showed that there is even less research on education, and it is particularly short in teacher education [19]. In higher education, research on gamification has focused on engineering and computers, which tend to focus on the use of technology and not the pedagogical side of the intervention programs [20]. Regarding Physical Education Teacher Education (PETE), [21] found that gamification increased students’ academic performance and external regulation. Ref. [22] uncovered that it promoted students’ motivation and commitment to the subject, although they preferred to work within homogeneous working groups. Ref. [23] highlighted that the students enjoyed the group dynamics, developed feelings of satisfaction and enjoyment, and increased their motivation and commitment. Ref. [24] revealed that their gamified program improved the students’ cardiorespiratory and physical health. Finally, in a study conducted by [25], participants acknowledged that they had learned to be better individuals and teachers thanks to the positive class climate created. 

Some authors believe that teaching is an emotional practice and consider assessing teacher’s emotions to understand interactions, identities, and values as something fundamental [26,27]. This becomes especially important in understanding how changes in pedagogy affect teacher educators and their students, who will also be future teachers [28]. Recently, [29] highlighted that “pedagogical change involved not only decisions and knowledge that are grounded in the practical and technical but also the emotional” (p. 11). They found that emotions are often paired (i.e., trust-distrust, uncertainty-confidence), which adds complexity. These authors “call upon others to share their insights… and establish… more complete understanding of the complex nature of PETE practice” (p. 12). This is very much needed when implementing new methodological approaches like gamification. 

Based on the previous information, and due to the lack of the literature related to gamification within PETE programs and the need for specific guidelines for practitioners, the following research question came up: “How do teachers and colleges evaluate the gamified experience in physical education teacher training?”. Taking this research question as a reference, and after designing and implementing a gamified experience at PETE, the objective of this study was to give voice to the participants (i.e., students and the teacher) to learn about their visions of the gamification experience. To achieve this aim, two specific goals were set: To assess the advantages and disadvantages of the framework from the students’ point of view.To evaluate the feelings and thoughts of the teacher on the design, implementation, and impact of gamification.

Finally, this paper also aimed to inspire teachers interested in applying gamification in similar contexts.

## 2. Materials and Methods

### 2.1. Context

This research was carried out during the course “Didactics of Physical Education II” (year three) of the Physical Activity and Sport Sciences degree from a private university in the northeastern region of Spain. 

### 2.2. Research Design

To fully understand the phenomenon under study, this research project followed a qualitative, descriptive method [30] and an action-research design [31]. The teacher-researcher wanted to improve his teaching regarding the design and implementation of gamification as a pedagogical framework in PETE [32]. Consequently, the action-research process followed the four-step cycle proposed by [33]: planning, acting, reflecting, and re-planning. In the third phase, the study incorporated the students’ voices.

### 2.3. Participants

One teacher-researcher (male, 36 years old) and the 74 students (86% males and 14% females, 22.8 ± 2.56 years) agreed to participate. The teacher-researcher (the first author of this study) adopted a full participation role. For the selection of the students, an intentional sampling was carried out [34]. The criteria for the selection of the participants were as follows: (a) enrollment in the course, and (b) complete follow-up and no dropout from the gamified experience. None of the students experienced gamification before, while the teacher-researcher could be considered an expert since he had knowledge of the framework and used it in previous years. He did not teach this group of students before.

### 2.4. Variables

There were three different types of variables: independent (i.e., the intervention program), dependent, and extraneous variables. 

#### 2.4.1. Independent Variable: Intervention Program

Following recommendations from [35], some of the most relevant characteristics of the proposed intervention are detailed below. As for the curricular elements of the gamified subject, Table 1 details its learning objectives and didactic contents. Regarding the intervention model and its context, the gamified intervention program was based on the comic book series “The Adventures of Asterix and Obelix: Asterix conquers Rome”. It was conducted during the second semester (from January to May) of the 2020/21 academic year: 18 weeks, 3 hours/week (1.5 h theoretical, 1.5 h practical). The intervention program was based on previous works on gamification [10,18], and it included the basic elements of gamification proposed by [9]. They are all synthesized in Table 1, and a few are shown in Figure 2.

#### 2.4.2. Dependent Variables

Students’ perceptions of gamification: advantages and limitations;Teacher’s perceptions of gamification: design and implementation; emotions, expectations, involvement, effects on students, and incidents.

#### 2.4.3. Extraneous Variables

With the selection criteria of the participants in this research, we wanted to limit the incidence of the following extraneous variables: (a) previous experience of the students in gamification, and (b) abandonment of gamification before its completion.

### 2.5. Data Collection

*Students’ open-ended questions*. At the end of the intervention program, all participating students were asked to answer two open-ended questions: (1) “What advantages would you highlight, if any, from the gamified experience?”; and (2) “What limitation did you find, if any?”. They were administered via an online questionnaire during the last class of the semester. Students answered using their electronic devices (i.e., mobile phones, laptops, tablets). Anonymity and confidentiality were guaranteed, so they were asked to answer honestly. These responses took up a total of 200 pages of a Microsoft Word document.

*Teacher’s diary*. During the implementation, the teacher-researcher completed a personal diary, where he reflected on the project on a day-to-day basis. He answered several questions: “What do you highlight from the last session?”, “What do you expect in the next one?”, “How did you feel before and after the session?”, “How do you think the students feel?”; “How are the sessions affecting the students?”; and “What incidents have occurred?”. The diary included a total of 60 pages of a Microsoft Word document.

### 2.6. Data Analysis

The results obtained from the students’ open-ended questions and the teacher’s diary were analyzed using thematic content analysis [36], constant comparison [37], and analytic induction [30]. The coding was carried out using NVivo12 software [38]. 

The key themes were coded as credible and reliable categories and subcategories [39], which also made it possible to identify useful citations. Several standards were followed to ensure methodological rigor (i.e., credibility, transferability, confirmability, and reflexivity) [40]:Credibility: persistent observation and data and analyst triangulation were used.Transferability: a rich, thick, and clear description of the whole process was utilized.Confirmability: the research team met regularly to discuss data analysis and interpretation.Reflexivity: all the mentioned standards were used to promote a reflexive climate.

#### Coding of Data Collection Instruments

To identify the text extracts related to the open-ended questions, pseudonyms were used together with their gender identification (male or female) (example: Luca-M). Regarding the diary, the texts were identified with the acronym (D) together with the number of the week from which the text originates (example: D-week four).

### 2.7. Procedure

First, the teacher-researcher designed the gamified experience. Second, permission to conduct the study was obtained from the Bioethics Commission of Universidad de Barcelona (protocol code 11/2021). Third, in the first class, the whole project was fully explained to the students, including data protection and confidentiality, and those willing to participate signed written informed consent. Only then, the intervention program explained in Section 2.4.1 started.

## 3. Results

### 3.1. Students

Data obtained from the students’ open-ended questions were grouped around two main categories: (1) positive: *framework, motivation,* and *transference*; and (2) negative: *boredom* and *group work* (Table 2).

#### 3.1.1. Framework

This was the strongest theme. Students mentioned the elements of gamification as the best part of the teaching-learning process that they had experienced. Students felt that “It was a very different way of conducting a subject” (Silvia-F), “A novel approach to connect theory and practice” (Miguel-M). Some even thought that “The university should introduce more gamified subjects” (Sofía-F). The framework used made the students want more:

“I wanted to know what was going to happen in the next class”(Luca-M)

However, they also acknowledged that it made them work hard:

“We worked harder than it looked; we learned almost without noticing it”(Clara-F)

The gamified framework helped to change the students’ views on the subject:

“At first, the subject did not look appealing, but the gamification made it worthwhile”(Jennifer-F)

“It is great to see that things are done differently, because it makes everything more attractive”(Víctor-M)

Enjoyment was a strong element in the framework because it included fun tasks and games:

“At the same time, you are learning and having super-fun”(Magdalena-F)

“I enjoyed it a lot. I have seen a new way of teaching through a parallel world [Asterix and Obelix]”(Francis-M)

Finally, the students mentioned the teacher’s role in the framework as a key element:

“I liked the idea and how it was conducted by the teacher”(Sofía-F)

“The teacher was absolutely involved in the class. It helped us become involved too”(Ricardo-M)

#### 3.1.2. Motivation

More than half of the students mentioned how the gamification was “An original idea that motivated students in their learning” (Maite-F) that had motivated them so much that “you wanted to go to class” (Patricia-F) and “make all the tasks” (Jaime-M). Elements like the narrative, the missions, the challenges, and the rewards were crucial for the success:

“From the beginning, it was motivating to be involved in a game based on Asterix and Obelix”(Héctor-M)

“I enjoyed trying to solve the challenges. It is exciting to reach different cities and move forward”(Antonio-M)

Working in groups and shifting the responsibility to the students were also important to foster their motivation:

“A new way of motivating students, a group bond was created, and cooperation improved”(Salva-M)

“It forces you to be alert every week and not to miss a thing. It is motivating, because it is not a subject where you sit, listen and take notes. Here you are the main character”(Silvia-F)

This boost in motivation also helped to enhance students’ commitment and connection with the teaching-learning process:

“Thanks to the gamification, I have been connected to the subject”(Antonio-M)

“It is a great way to motivate and stimulate students to go to class; specially to theory, where attendance is always low. Without gamification, I doubt this class would be interesting”(Carlos-M)

Commitment also increased towards group work thanks to different elements (i.e., challenges, rewards):

“Gamification is a good way of working. It made those students disengaged from college develop feelings of belonging to their group and not fail”(Ricardo-M)

“This experience has made me become more engaged in the lessons; specially during the tasks to solve the challenges and earn the rewards”(Luca-M)

Moreover, motivation and commitment helped students learn:

“I learned more and better”(Sandra-F)

“Learning was meaningful”(Miguel-M)

“I learned more because I became more involved”(Aura-F)

#### 3.1.3. Transference

A small part of the students mentioned that the gamified framework could work well in primary and secondary physical education because “our school system must be upgraded” (Héctor-M), and “gamification can be beneficial for children, since it has been a memorable experience” (Marta-F).

#### 3.1.4. Boredom

A few comments mentioned that the experienced was tiring and, for some students, the whole framework was complex:

“I learned because I became involved, but at times it was wearing”(David-M)

“The problem was that it was a bit complex. Too much information”(Estefania-F)

“At times, I felt lost”(Salva-M)

Some students also complained about the childlike narrative and the routines to obtain the rewards:

“It can be boring, because we are adults, and the narrative was not real”(Enrique-M)

“It was burdensome when we had to review all the shields and the points on each village”(Lara-F)

#### 3.1.5. Group Work

It could be considered a minor theme because only two comments emerged. Students complained about the heterogeneous grouping because the groups did not function properly and it was frustrating because it increased their workload:

“The groupwork was not always cooperative”(Marina-F)

“This framework wanted us to work cooperatively. It was not achieved in my group. Not everyone had the same interest, and, at times, one part of the group became frustrated because they finished the challenges, while others did not worry”(Flora-F)

#### 3.1.6. Summary

The elements of gamification (dynamics, mechanics, and components), together with the fun generated and the role of the teacher, were crucial for increasing student motivation and commitment to the subject, as well as their learning. For this reason, the students highlighted the importance of transferring gamification to today’s schools. However, a small group of students reported feeling bored during the course, while others complained about the heterogeneity of the groups.

### 3.2. Teacher

Data obtained from the teacher’s diary were also grouped around two main categories: (1) positive: *mixed emotions, expectations, and students’ motivation*; and (2) negative: *workload* (Table 3).

#### 3.2.1. Mixed Emotions

This theme addressed the impact of the gamified experience on the teacher but also on the students. Regarding the former, it included a contradictory emotional state since the teacher navigated between two ends: joy-happiness and sadness-fear. Prior to every session, the teacher was “happy and joyful” (D-week three), caused by some elements of the gamified framework:

“I still have butterflies in my stomach. Students are going to love the cards, the shields, and the board. I am optimistic”(D-week two)

“I believe that the Breakout-Edu is going to work fine. I want to begin because I am happy with the work done”(D-week 15)

From the beginning, the teacher was ready for action:

“I want to go to work. I want to see the students’ faces during the special event”(D-week five)

“I want the Monday class to begin”(D-week six)

However, there was also tension. The teacher was “afraid of the students’ reactions when explaining the groups’ formation, because they had to be heterogeneous in gender and grades” (D-week one). He also felt “worried about problems within the groups, especially when some members did not fulfill the agreements” (D-week five), struggling when the students did not achieve the expectations:

“Another group did not submit the project. It is the second one”(D-week four)

This caused tension in the teams since “some students showed their discomfort with some group members because they were not involved in the teams’ work” (D-week 14). Some students even wanted to move to another group: 

“Today, I received an e-mail from a student who wanted a group change. His group has not turned the assignment”(D-week four)

In line with this issue, “the apathetic attitude of a couple of students in class” (D-week three) was also troublesome because it made the teacher uncomfortable. In the same way, but even more significant, were the comments from another university teacher, which made the former feel “very upset, because the comments were made in front of the students, taking away [his] credibility” (D-week four):

“He has made some critics of the board exposed in the class: it lacks rigor and gamification too”(D-week four)

The framework’s effectiveness also caused some negative feelings:

“I have doubts about the usefulness of gamification its impact and the students’ commitment”(D-week 17)

“[I feel] doubtful I don´t know if all this really works. I don´t know if students have learned”(D-week 14)

In fact, after correcting some tasks on theoretical content, the teacher felt “sad and sorry, because the essays are not as good as they should be” (D-week four). Furthermore, he felt “worried because I don’t not know if the students integrated only the playful anecdotes and not the real content” (D-week eight). This emotional state was kept throughout the subject:

“I saw high motivation in the class, but when students’ cognitive and academic implication is demanded, I see problems. I don’t know if I will be able to achieve my goals”(D-week seven)

Nevertheless, the teacher was happy with the final grades, because “only four students failed, which is 5% of the class. The majority integrated the subject’s main contents” (D-week 18).

Regarding the students’ feelings, the teacher observed mixed emotions. When the narrative was introduced “some were very surprised and attentive to all the new things, while others were indifferent” (D-week two). However, the teacher highlighted that one of the groups “has reacted very lively to the narrative… clapping, cheering and laughing” (D-week two), and “they left the class saying goodbye with joy” (D-week two). Most comments reflected students’ happiness during the semester, especially during the special events and the rewards:

“Giving points and moving around the board was great. Students looked happy, particularly when they earned some pirate coins”(D-week three)

“Students were super-happy. Lots of laughs and interest to solve the enigmas”(D-week seven)

“Students were excited with the scratch cards. One told me that he was happy and proud to be able to solve the task”(D-week 15)

Unfortunately, frustration also emerged “when one student did not obtain the desired score, he had to give out some coins to try again. A couple of students complained” (D-week four).

Finally, since feedback and corrections on the tasks and essays were constant, the teacher did not observe “fear to the assessment… on the contrary, I see interest to improve for the following week” (D-week five), even though “some groups were frustrated when their grade was not high enough, but the possibility to make the task again, to improve the score, helped them calm down” (D-week eight).

#### 3.2.2. Expectations

The teacher’s biggest expectation was linked to students’ commitment. During lesson design, the teacher showed high interest to “help students to maintain or increase their participation” (D-week one) towards the subject and the gamified tasks. To achieve this goal, the basic elements of gamification (i.e. dynamics, mechanics, components) were very relevant:

“I dramatized while explaining the first mission, because I want to get the students motivated and committed from the beginning”(D-week two)

“With the shields’ special event I want to motivate students to perform the extra tasks”(D-week five)

“I hope that all the students that still can obtain the scratch cards perform the needed tasks”(D-week 15)

“I am eager to begin the Breakout-Edu… it is the star session, [it required] lots of preparation.”(D-week 16)

As the sessions went by, the teacher also hoped to generate positive reactions. The basic elements (i.e., dynamics, mechanics, components) were again crucial:

“Tomorrow, they [the students] will be speechless [with the narrative]. I will bring a box with all the costumes. I hope they like them”(D-week one)

“This week can be fun because the dynamics are playful: move over the board, collect pirate coins, shields…”(D-week three)

The teacher hoped “to hook the students to the gamification” (D-week five). However, he was worried about keeping the students permanently connected (flow) to the gamified experience, and used the special events because “they are going to make an impact and help the students become connected to the gamification” (D-week seven). The teacher also expected to increase the students’ motivation “to the coins, the passport, the mission or the victory over Brutus [elements]” (D-week four) or “to the routes and the election of the right path [elements]” (D-week five). The goal was to increase students’ learning during the different sessions and “help all the students pass the final exam” (D-week 17).

#### 3.2.3. Student Motivation

It was present from the beginning; when the narrative was introduced “some students’ faces lighted up and they [students] could not stop smiling” (D-week one). Earning rewards helped stimulate the students:

“I saw high motivation to win the coins”(D-week four)

“They [students] constantly ask about the possibility of obtaining more coins or win back the lost ones”(D-week seven)

“Groups designed strategies to earn more pirate coins next week”(D-week three)

“More than 50% of the students are doing the extra tasks [to earn coins]”(D-week nine)

This high motivation also led to other elements such as effort and participation, which helped to create “a good climate” (D-week eight):

“Students tried to make the challenges”(D-week four)

“The students are highly involved in class, they ask questions, review the difficult ideas”(D-week six)

The special events were also important since “students are motivated; the uncertainty that they [special events] generated, left students speechless” (D-week 16). The teacher believed that “both class groups show a high degree of involvement towards the subject and the gamification” (D-week six).

#### 3.2.4. Workload

The teacher considered that the design and implementation of the gamification increased his workload. Gathering the resources needed for the narrative demanded a lot of time in the beginning:

“Like last week, it is the day prior to the class, and here I am preparing stickers and passports”(D-week two)

“As usual, it is Sunday, and I am still working on the board”(D-week three)

The special events also increased the workload because “they need a lot of time to plan and design until the last minute” (D-week three). Even the final event demanded “a lot of work” (D-week 17) during several weeks that included some training:

“I found extra information on Edu-Breakouts in a seminar, and I attended it to use it in class”(D-week 14)

The increased workload was also distressing because of the lack of some resources:

“I ran out of coins, and I did not expect it. I should have bought more, but I couldn’t because of the busy week. The groups couldn´t take home all the coins that they earned”(D-week eight)

Providing feedback during tasks and assessments also increased the workload. Moreover, students were allowed to repeat an assignment with a low grade, which also increased the workload:

“This week’s session is important, because I am going to provide feedback on the first mission challenges”(D-week four)

“The students turned in the challenges of the second mission. I must correct them all before the next class to offer guidance”(D-week 13)

“This week I have to assess the assignments of the groups that did not pass the first time”(D-week five)

Finally, the teacher-student interaction outside the class also increased. Through Classdojo, the teacher frequently contacted the students:

“Today, I sent a message to the students pretending to be Julius Caesar. I told them that they had obtained the passport to enter Britain”(D-week four)

“I wanted to motivate the students and to advertise the special event. I sent them a message”(D-week seven)

#### 3.2.5. Summary

The teacher expressed a contradictory emotional state, which was positive in terms of the development and preparation of the sessions, but negative when faced with doubts about their appropriateness or the real impact on the students. Most of the teacher’s expectations were related to increasing student engagement, learning, and especially motivation. Therefore, the teacher was concerned about using the game elements appropriately to maintain the flow towards the gamified experience. Through his diary, the teacher reported a high increase in his workload as a negative aspect. This was due to the preparation of the materials for the gamified experience itself and the constant feedback that the teacher gave to the students about their challenges.

## 4. Discussion

The aim of the present study was to obtain the participants’ opinions about the gamified experience, and the results were very constructive. From the students’ responses emerged three positive themes (i.e., framework, motivation, and transference) and two negatives (i.e., boredom and group work), and from the teacher appeared three positive (i.e., mixed emotions, expectations, and student motivation) and one negative response (i.e., workload) (Figure 3).

The first goal was to assess the advantages and disadvantages of the framework from the students’ point of view, and the results showed that the advantages were superior. The framework, which included all the elements of gamification, was the best advantage of the intervention program. It helped students “look at old things in new ways” and “do new things in new ways” from a transformative learning perspective [4]. The gamified framework forced students to critically reflect on their learning, engage in productive dialogue within their working groups, and live individual first-person experiences throughout the semester inside a motivating context [5]. Based on their previous experiences, these university students found the gamified experience novel and innovative, which was able to feed their curiosity and uncertainty within a positive emotional climate. Previous research found that these ingredients are needed for gamification to be effective [14]. These results are in line with previous programs conducted in PETE programs [22] where all participants enjoyed the gamified experience, which is a key element for a successful implementation of gamification [9]. The novelty has been linked to enjoyment in education [41], and the results from the present study confirmed it since the participating students enjoyed the intervention program. In this framework, the teacher’s role and his commitment were also highlighted. Both have been considered key elements in the successful integration of gamification in education [42], though sometimes, they are underrated by those teachers who want to use this framework. The results from the present study indicate that university teachers need to be active participants in their instructional frameworks if they want to make a significant impact on their students [32].

The second group of positive views on the experience was the encouraging effects of the intervention program. Many students believed that gamification had increased their motivation in the subject, due to the game elements used, such as the narrative, the missions, the challenges, and the rewards. Previous research also found that these elements play a key role in promoting students’ motivation [21]. Other elements such as group work or an increase in the students’ responsibility in class were also considered important. Although this study did not assess different types of motivation, students’ comments showed that both (i.e., intrinsic and extrinsic) were present: intrinsic motivation linked to novelty (narrative, challenges) [43], autonomy, individual progress, and social relations [44], as well as extrinsic motivation linked to rewards. To minimize the negative effects of external rewards on individuals’ intrinsic motivation and in line with previous studies [43], rewards were immediate and there were no public scoreboards that favor competition and comparison between groups. Previous research on college students’ intrinsic motivation after experiencing gamification showed contradictory results: [18] found a significant decrease while [22] found an increase. 

More research is needed to understand this new pedagogical framework. In the present study, the increase in motivation probably caused an increase in students’ commitment and involvement, which also seemed to favor learning. This should be the true goal of gamification in educational contexts: achieving education-related goals and learning, not just enjoyment [45]. These outcomes are the perfect examples of the behavioral changes that gamification tries to induce in any participant [9], but this is particularly important in teacher education because participants will carry these changes into their future professional practice. 

The third and final positive outcome from the students’ point of view was transference. This group of future teachers believed that gamification is a “good tool” for their professional future because it will “work fine” in primary and secondary education. This could be considered noteworthy and encouraging since first-hand experience is usually used later in life. This positive view could help to expand this framework, shift the focus from the teacher to the students, and create student-centered educational contexts [6] to provide successful experiences that can fulfill all students’ needs. Teacher education has the “duty” to provide transformative learning experiences for future physical education teachers [3], which they could use in their professional practice, and the results from the present study confirmed that the gamified framework achieved this goal. Teacher education must play a proactive role to assess and integrate new methodological approaches such as gamification that can bring education closer to 21st-century students.

Regarding the negative outcomes, a small number of students “felt lost” during the experience, which led them to think that it was, at times, “monotonous and boring”. Previous research is aligned with these results, showing that college students tend to resist methodological changes because of “the dead load of their educational experience” [24] (p. 258). Experiencing gamification during just one semester and one subject could not be enough to change some college students’ thoughts. Therefore, any teacher in teacher education willing to conduct a gamified experience must be aware of different existing student-player profiles to meet their needs. 

In this same line, inadequate group work was also pointed out as a limitation, results that are in line with previous research [22], since some students complained about heterogeneous grouping and the “non-commitment” of some members. This is a sign of the lack of cooperative spirit among some college students [25], which could come from a limited experience in cooperative learning in previous educational stages.

The second goal of the study was to assess the feelings and thoughts of the teacher on the design, implementation, and impact of gamification in teacher education, and the results were very positive. Changing the pedagogical approach is no easy task for teachers. It is important to understand how this process emotionally affects teacher educators and their students, future teachers [28], to help other educators face that challenge. 

Mixed emotions were the strongest theme, and it included ambivalent feelings, ranging from joy-happiness to sadness-fear. Ref. [29] found that emotions come often paired (i.e., trust-distrust, uncertainty-confidence), which adds complexity to the gamification process [28]. In the present study, positive emotions appeared while the teacher was preparing different resources for use during the implementation, while negative ones were linked to the program’s effectiveness. He felt “sad and sorry” when his expectations were not fulfilled. This is important because, to the knowledge of the authors, this is the first time that gamification has been assessed from the teacher’s point of view, and these results clearly show that an adequate implementation is not an easy task, and many teachers do not go beyond the “honeymoon period” [46] and quit when they try to use a new pedagogical approach. 

Regarding the participating students’ emotions, the teacher’s comments reflected joy and happiness, which could be considered normal when there are rewards and special events. The diary showed that the class climate was positive and that students enjoyed the lessons and had fun, something previously observed in gamification [10]. The teachers’ comments also reflected a decrease in the students’ fear of assessment, which is in line with previous studies that showed that game-like dynamics can reduce students’ fear to make mistakes [22].

Although gamification has several benefits, the implementation of a gamified subject significantly increased the teacher’s workload. The design and preparation of all required materials and special events took a lot of time, mostly out of working hours. The use of formative assessment, which included the possibility to resubmit assignments, as well as the teacher-student interaction out of class through messages in Classdojo to keep students’ attention also increased the workload. Previous research also described an increased workload that diminished over time [47], which was not the case here, probably because college students demand more, long-lasting attention. According to [48], the increased workload is the main cause of desertion when trying to implement new pedagogical approaches.

Finally, the teacher’s comments also reflected his expectations: increase students’ commitment and flow in an emotionally positive and fun class climate to increase their motivation towards learning the subject. He hoped that these expectations would turn into reality with the final grades. To achieve this goal, the basic elements of gamification (i.e., dynamics, mechanics, and components) were fundamental. As previously mentioned, these high expectations generated fear and, at times, sadness in the teacher. Fortunately, the students acknowledged that their motivation and commitment to the subject increased, which showed that the expectations of the teacher were real. Moreover, the diary also reflected this increase: students “tried hard” to conquer all the challenges and obtain all the rewards, which showed in their final grades (they were good). This enhancement of students’ motivation and commitment was described in previous studies on college students [23]. The teacher’s efforts to maintain a balance between enjoyment, motivation, and learning were crucial for the success of the experience because they are considered key elements for the effective implementation of gamification [42].

All things considered, gamification seems to be an adequate approach to achieving the fourth Sustainable Development Goal established in the 2030 Agenda by being a pedagogical renewal of traditional teacher education and improving the teaching-learning process [1,2,3]. Gamification can provide transformative learning experiences to future teachers, enhancing and increasing their pedagogical repertoire and, in the end, having a positive impact on their future pupils and society.

This work has several practical implications that should be noted. On the one hand, it is useful for putting into perspective the positive and negative effects of the application of a gamification in the teaching-learning process based on the voices of its principal agents. On the other hand, for teachers who consider applying gamification in their teaching, it provides certain guidelines on how to apply a standard proposal in the field of physical education. In this sense, the work presented clearly shows the difficulties and benefits that an intervention of this type implies for the person who puts it into practice, in a way that allows the person who wants to carry it out to learn from the successes and mistakes of other researchers. Thus, it is essential for any practitioner who wants to design a gamified teaching-learning experience to bear in mind the importance of reading research such as the present study, which may allow them to identify the key features of this type of innovative methodologies.

## 5. Conclusions

The use of gamification in PETE has been perceived as positive by the participants (i.e., students and the teacher), which showed its feasibility and suitability in this educational stage. The students believed that the framework turned the subject into a novel, original, and surprising experience, which promoted enjoyment thanks, among other things, to the teacher’s role. The positive climate generated improved students’ motivation and commitment to the subject, which favored learning. These outcomes matched the teacher’s expectations and were also perceived and identified by the teacher. Based on the aforementioned, gamification could be considered a framework that promotes transformative learning. 

However, a few limitations need to be considered in the present study. Some students felt bored and tired, and their groups did not work properly; the teacher believed that gamification impacted him emotionally from two ambivalent perspectives: joy-happiness and sadness-fear. Changes in pedagogy are no easy task, and they can emotionally affect educators and their students. Other limitations were an increase in the teacher’s workload, the limited sample size, the fact that only qualitative analyses have been carried out, the limitation of the existing literature on gamification, and physical education teacher training, especially from a qualitative perspective. 

In order to mitigate these limitations, future research should address the effects of gamification on students and, especially, on teachers, since these have often gone unnoticed in the existing research. Moreover, further studies could carry out mixed-method designs, combining qualitative and quantitative research and recruiting a larger sample to gain a deeper insight into the effects and consequences of gamification, as well as establishing methodological procedures and frameworks to reduce the workload of teachers willing to use gamification in their classes. 

## Figures and Tables

**Figure 1 ijerph-20-03038-f001:**
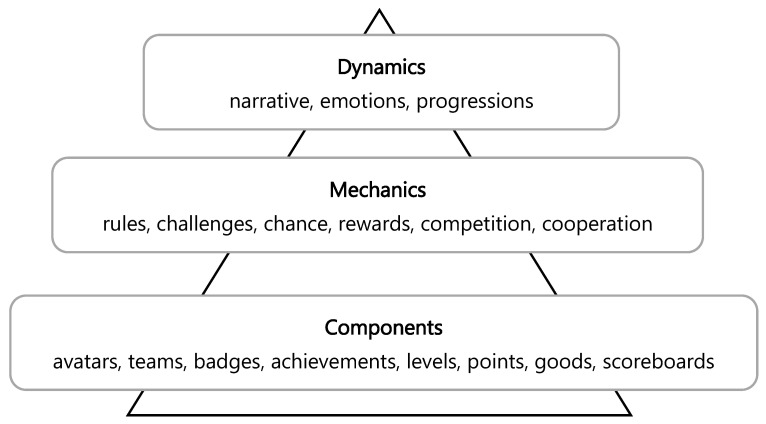
Game design elements. Adapted from Werbach and Hunter [9].

**Figure 2 ijerph-20-03038-f002:**
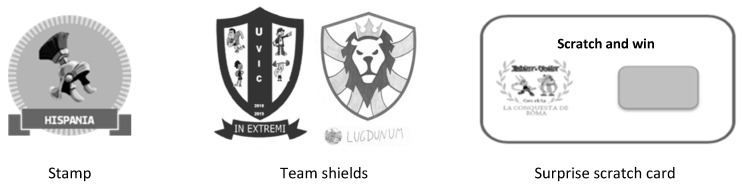
Game elements.

**Figure 3 ijerph-20-03038-f003:**
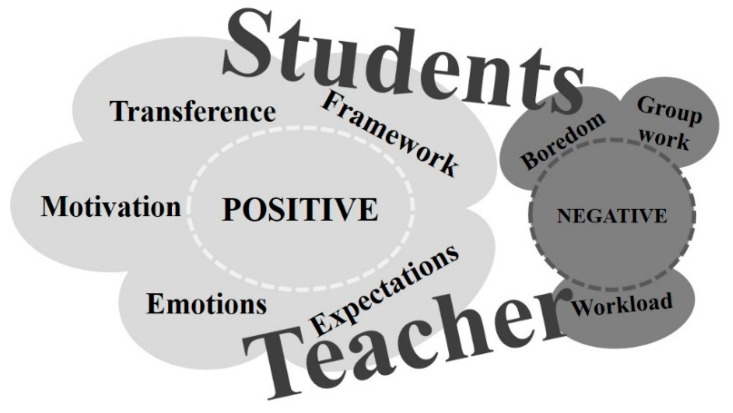
Results of the thematic content analysis.

**Table 1 ijerph-20-03038-t001:** Elements used in the gamification “Asterix conquers Rome”.

**NARRATIVE** Inspired by the comic book series “The Adventures of Asterix and Obelix”, it described the defeat of Julius Caesar. The final goal of the project was to travel to Rome to defeat Julius Caesar and the Roman Empire.
**MISSIONS and CHALLENGES** Matching the subject’s themes/topics, the trip to Rome included four missions: (1) *the trip to Britain:* 21st-century physical education;(2) *the trip to Hispania:* assessment in physical education; (3) *the trip to Athens:* didactic bases for the preparation, management, and evaluation of the practice; and (4) *the arrival in Rome:* innovative pedagogical methodologies. Each mission included theoretical challenges (19), which helped the students achieve the subject’s final outcomes. Completing each mission was mandatory to continue the trip and reach each destination, where they had to defeat a Roman General.
**PLAYERS and TEAMS** The teacher played Julius Caesar and each of the Roman Generals. Nineteen heterogeneous groups of four members were created based on gender and grade point average. Each group represented a Gaul village, and each student performed one role of the story: Asterix, Obelix, Getafix, and Vitalstatistix. Each group designed its own shield, which identified it during the whole adventure (avatar). Everyone signed a written contract to work as a group.
**REWARDS** *Magic drink*: Students earned them at the end of every challenge; every task yielded a different number of “drinks”, and all missions required a minimum number of “drinks”. If one group did not earn enough drinks, they were granted extra time to solve the challenges (correct the assignments). The drinks were essential to “fight the Roman Generals”. If one group earned the required number of drinks, they could “roll the dice” to obtain “more than two points” and fight. If they got a “one”, they had “to pay a coin” to roll again. *Coins*: The students could earn coins scattered throughout the map. Coins could enhance the teams’ strength in the final event (Breakout-Edu), but they could also be used to earn resources or traded between teams.
**GOODS** *Passport and stamps*: To be able to travel (missions), students had to complete the group’s passport. Each country’s stamps were obtained after defeating each Roman General. To reach the final mission (Rome), it was mandatory to collect all the stamps.
**POINTS OF EXPERIENCE** The *defensive shields* were obtained after successfully completing extra tasks. Three shields gave the group a *surprise scratch card*, which included benefits to be used during the final written examination, such as extra points, the use of their notes, the possibility to change one of the questions, etc.
**GAME BOARD** It was a big map located in the class. Each group had a game piece with its shield, which was moved to different destinations along three paths. Displacement was dependent on the points earned weekly (previously agreed with the teacher). The points were registered using *ClassDojo.* The board did not highlight winners or losers; it just helped every team locate itself during the game.
**SPECIAL EVENTS** There were five events: (1) campus search for defensive shields; (2) Kin-ball season and championship; (3) winter has come: session frozen and blocked; (4) the villages’ rebellion: fight between teams; and (5) Breakout-Edu: “catch Julius Caesar”.
**AWARDS** During the last session, awards (self-constructed trophies) and diplomas were handed to all the participating teams.

**Table 2 ijerph-20-03038-t002:** Themes and meaningful segments from students.

**Positive**	
**Themes**	**Meaningful Segments**
Framework	61 (53%)
Motivation	49 (41%)
Transference	8 (6%)
**Negative**	
**Themes**	**Meaningful Segments**
Boredom	9 (81%)
Group work	2 (18%)

**Table 3 ijerph-20-03038-t003:** Themes and meaningful segments from the teacher.

**Positive**	
**Themes**	**Meaningful Segments**
Mixed emotions	29 (37%)
Expectations	27 (34%)
Student motivation	12 (15%)
**Negative**	
**Themes**	**Meaningful Segments**
Workload	11 (14%)

## Data Availability

The data presented in this study are available on request from the corresponding author. The data are not publicly available due to privacy restrictions.

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
