# Peer review of "“I Learned More Because I Became More Involved”: Teacher’s and Students’ Voice on Gamification in Physical Education Teacher Education"

_ijerph, 2023, doi:10.3390/ijerph20043038_

Round 1

Reviewer 1 Report

Dear authors, 

First of all, congratulations on your work. Broadly speaking, the subject matter seems to me to be interesting and up-to-date, in need of this type of research in response to new educational trends. Here are some suggestions that can improve the quality of your contribution:

-I would recommend revising the wording of the text with regard to the quotations. In particular, I sometimes get the impression that the text was written according to a specific citation rule that was subsequently changed/adapted, with certain parts of the text cutting off the reading.

- Perhaps within the method it would be interesting to include the role of the teacher who led the gamification experience with respect to the development of this research and the analysis of the results. 

Kind regards.

Author Response

Dear reviewer,

Thank you for your kind words and your suggestion about this research, we are glad to hear that other experienced researchers find this study useful.

Regarding your recommendation, we have modified the structure of the results section and we have formatted the quotations following MDPI’s citation style (lines 235-442), and the role of the teacher with respect to the research has been included (lines151-152). Lines are numbered according to the track changes option “show all changes”.

Again, thank you very much for your time and effort in reviewing our manuscript. Best regards.

Reviewer 2 Report

Dear authors, first of all thank you for submitting to IJERPH.

The submitted article assumes enormous relevance in the current spectrum. In fact, gamification is currently one of the topics that most needs to be explored, taking into account its potential applicability in the context of Physical Education, via the associated increase in motivation.

The authors have done a good job and deserve credit for it, yet there is one specific point I would like to see improved.

It would make perfect sense for the authors to present at the end a topic with potential practical applications/recommendations derived from their research. It is critical to increasing the value of the manuscript.

Overall good job.

Author Response

Dear reviewer,

Thank you for your kind words and your suggestion about this research, we are glad to hear that other experienced researchers find this study useful.

Regarding your recommendation, we have included a paragraph about potential practical applications and recommendations (lines 584-595). Lines are numbered according to the track changes option “show all changes”.

Again, thank you very much for your time and effort in reviewing our manuscript. Best regards.